# Synthetic Cationic Lipopeptide Can Effectively Treat Mouse Mastitis Caused by *Staphylococcus aureus*

**DOI:** 10.3390/biomedicines11041188

**Published:** 2023-04-17

**Authors:** Jie Peng, Qiangsheng Lu, Lvfeng Yuan, Hecheng Zhang

**Affiliations:** 1College of Veterinary Medicine, Gansu Agricultural University, Lanzhou 730070, China; 2Lanzhou Veterinary Research Institute, Chinese Academy of Agricultural Sciences, Lanzhou 730046, China; 3State Key Laboratory of Veterinary Etiological Biology, College of Veterinary Medicine, Lanzhou University, Lanzhou 730000, China

**Keywords:** ultra-short cationic lipopeptides, *Staphylococcus aureus*, mice, mastitis

## Abstract

Mastitis caused by *Staphylococcus aureus* (*S. aureus*) in dairy cows is one of the most common clinical diseases in dairy cattle. Unfortunately, traditional antibiotic treatment has resulted in the emergence of drug-resistant strains of bacteria, making this disease more difficult to treat. Therefore, novel lipopeptide antibiotics are becoming increasingly important in treating bacterial diseases, and developing novel antibiotics is critical in controlling mastitis in dairy cows. We designed and synthesized three cationic lipopeptides with palmitic acid, all with two positive charges and dextral amino acids. The lipopeptides’ antibacterial activity against *S. aureus* was determined using MIC and scanning electron microscopy. The safety concentration range of lipopeptides for clinical usage was then estimated using the mouse erythrocyte hemolysis assay and CCK8 cytotoxicity. Finally, lipopeptides with high antibacterial activity and minimal cytotoxicity were selected for the treatment experiments regarding mastitis in mice. The observation of histopathological changes, bacterial tissue load and expression of inflammatory factors determined the therapeutic effects of lipopeptides on mastitis in mice. The results showed that all three lipopeptides displayed some antibacterial activity against *S. aureus*, with C16dKdK having a strong antibacterial impact and being able to treat the mastitis induced by *S. aureus* infection in mice within a safe concentration range. The findings of this study can be used as a starting point for the development of new medications for the treatment of mastitis in dairy cows.

## 1. Introduction

*Staphylococcus aureus* (*S. aureus*) is a common food-borne gram-positive pathogen that is frequently isolated in cases of bacterial mastitis in dairy cows. This is the major pathogen causing mastitis in dairy cows in several regions [1]. Mastitis in cows causes redness, swelling, fever, soreness, and other symptoms in the mammary gland. When milk production is diminished, the quality of milk suffers, and some sick cows do not produce milk. Most *S. aureus*-caused cow mastitis cases are difficult to treat and generate significant economic losses [2,3]. In addition, *S. aureus* is responsible for the development of major septic and toxigenic illnesses, in addition to mastitis. Meanwhile, *S. aureus* is also an important class of human–animal pathogens that can cause a variety of infections, including skin infections, pneumonia, and endocarditis. In addition, it can also cause food poisoning [4]. Therefore, it has very important public health significance.

The current treatment of *S. aureus* mastitis in dairy cows relies on traditional antibiotics. However, the misuse of antibiotics has led to many severe problems, such as the development of bacterial resistance. In addition, public health problems, such as drug residues in beef and cow’s milk caused by antibiotic abuse and the resulting development of super-resistant “superbugs”, are also serious health risks [5]. As *S. aureus* produces penicillinase or carries genes for tetracycline and erythromycin resistance, they are more likely to develop antibiotic resistance. As a result, *S. aureus* is of critical public health importance. Lipopeptides have attracted significant attention in recent years for their novel mode of action and broad-spectrum antimicrobial activity against various pathogens. Conventional antibiotics generally target metabolic enzymes and can selectively produce resistance. However, due to their lipophilic activity, lipopeptides can kill bacteria by disrupting their cell membranes and causing damage that is difficult to repair, making them less likely to develop resistance compared to traditional antibiotics [6]. Therefore, lipopeptides have great potential to replace traditional antibiotics [7,8,9].

Numerous studies have revealed that a positive charge and hydrophobicity are critical to antibacterial peptide biological activity and mechanisms. Essential amino acids, such as lysine and histidine, often provide a positive charge. The combination of fatty acids with proteins or peptides considerably increases hydrophobicity and promotes membrane contact considerably [8,9,10]. The creation of a novel class of short cationic lipopeptides involves connecting dipeptides, tripeptides and tetrapeptides to palmitoyl chains via their N-terminus [9]. The selection of amino acids on the lipopeptides fulfills the minimal range of functions necessary for successful antibacterial activity, and the utilization of this class of antimicrobial peptides has the benefit of reduced manufacturing costs and shorter synthesis times [11]. The case of chemical surfactants shows that the longer the lipophilic acyl chain, the more effective the micellization due to an increase in intermolecular hydrophobic interactions [12]. Antimicrobial activity studies have shown that lipopeptides with C16 alkyl as fatty acids have substantial antibacterial effects in antimicrobial activity studies [6].

Three lipopeptides were developed and produced in this study. The short peptide’s essential amino acid is dextro-lysine, with two, one, or no glycine inserted between the two crucial hydrophilic amino acids. Palmitic acid (C16) was chosen as the fatty acid. The effects of the composition and arrangement of amino acids on their antimicrobial activity and cell selectivity were studied systematically. Subsequently, the best drug candidates were evaluated for their therapeutic efficacy in *S. aureus*-caused mouse mastitis to demonstrate their clinical application. Finally, we hope to explore whether related lipopeptides can effectively treat mastitis caused by *Staphylococcus aureus* in mice through this study.

## 2. Materials and Methods

### 2.1. Strains

*S. aureus* strain ATCC 25923 was gifted by the Laboratory of Microbiology and Immunology, Nanjing Agricultural University, and kept by the Laboratory of Microbiology and Immunology, Gansu Agricultural University. The *S. aureus* strain GS1311 was isolated from the milk of a clinically ill cow on a dairy farm in Gansu Province, China. The strain was isolated and preserved in our laboratory. Bacterial passaging cultures were cultured overnight at 37 °C in a tryptone soy broth (TSB) medium (Becton, Franklin Lakes, NJ, USA).

### 2.2. Animals and Cells

Kunming pregnant mice were purchased from the Lanzhou Veterinary Research Institute of the Chinese Academy of Agricultural Sciences. All animal experiments were performed with protocols approved by the Lanzhou Veterinary Research Institute for Research Protection Standing Committee on Animals in accordance with the Science and Technology Agency of Gansu Province guidelines (SYXK20200010). The mouse mammary epithelial cells EPH4-Ev were purchased from American Type Culture Collection (ATCC number CRL3063) and kept in our laboratory. Cells were cultured in DEME (Gibco, Grand Island, NY, USA) containing 10% fetal bovine serum (FBS) (ScienCell Research Laboratories, San Diego, CA, USA).

### 2.3. Lipopeptide Synthesis

All three lipopeptides involved in this study were synthesized by Suzhou Yao Qiang Biotechnology Co., Ltd (Suzhou, China), and the products were available in the form of lyophilized powder. The synthesized lipopeptides were C16dKdK, C16dKGdK and C16dKGGdK (Table 1). The purity and molecular weight of the synthesized products were examined by high-performance liquid chromatography and mass spectrometry. The lipopeptides were prepared in sterile PBS with a final concentration of 10 mg/mL storage solution and stored at −80 °C for backup.

### 2.4. Determination of MICs

The minimum inhibitory concentrations (MICs) of three peptides were measured by the micro broth dilution method to evaluate the antimicrobial activity of new derivative peptides [13]. In brief, the test strains were cultured in TSB medium at 37 °C until the mid-log phase and then diluted to 5 × 10^5^ CFU/mL. The peptide solutions were diluted twofold with the final concentrations of 0.05–400 μg/mL. A total of 10 μL of peptide solutions and 90 μL of exponential phase bacterial suspension were added to 96-well microtiter plates and incubated at 37 °C for 18–24 h. All assays were performed in triplicate. MIC was defined as the lowest concentration with no visible bacteria growth after overnight incubation.

### 2.5. Electron Microscopy Observation

Electron microscopy was used to observe morphological changes in cells after treatment with the peptides. The method referred to Liu et al. 2021 [14] but was slightly different. In brief, the mid-log phase *S. aureus* ATCC25923 and *S. aureus* GS1311 cells (1 × 10^8^ CFU/mL) were incubated with peptides (C16dKdK, C16dKGdK and C16dKGGdK) at a final concentration of 2 × MIC at 37 °C for 2 h. The cells were centrifuged at 4000 rpm for 5 min, washed with PBS three times, and fixed in 2.5% glutaraldehyde overnight at 4 °C. For scanning electron microscope (SEM), the cells were washed with PBS, dehydrated with a graded ethanol series (50, 70, 80, 90, and 100%) for 10 min, and dried by CO_2_ using a Critical Point Dryer (Automegasamdri-938, Series C Tousimis, Rockville, MD, USA). The specimens were coated with gold/palladium, and the images were captured using SU8100 (HITACHI, Tokyo, Japan).

### 2.6. Hemolytic Assay

The hemolytic activity was determined according to the method of Yuan Lvfeng [15]. The mice’s blood cells were collected and washed three times with 0.9% NaCl. One hundred microliters of 2% (*v*/*v*) erythrocyte solution were mixed with 100 μL of peptides (C16dKdK, C16dKGdK and C16dKGGdK) to the final concentration of 0.2–400 μg/mL. After incubation at 37 °C for 1 h, the mixture was centrifuged at 1000 rpm for 10 min at 4 °C. A total of 100 μL of supernatant was transferred to a new microtube and measured at 540 nm(OD_S_). Values of 0.9% NaCl (OD_B_) and 0.1% Triton X-100 (OD_P_) were used as controls. The hemolysis percentages of the peptide were calculated using the following equation: Hemolysis (%) = [(OD_S_ – OD_B_)/(OD_P_ – OD_B_)] ×100%.

### 2.7. Cytotoxicity Assay

Mouse mammary epithelial cells EPH4-Ev cell line was used to determine the cytotoxicity of the peptide by CCK8 assay. In brief, cells were inoculated into 96-well cell plates at a density of 1.0 × 10^5^ cells/well and interacted with various peptides (0.1–200 μg/mL) for 2 h at 37 °C and 5% CO_2_. Subsequently, the cell cultures were further incubated with 10 μL of CCK-8 detection reagent (Shanghai Biyuntian Biotechnology Co., Ltd. (Shanghai, China)) for 1 h at 37 °C. The solution was further measured using a SpectraMax 13× (Molecular Devices, San Jose, CA, USA) at an optical density (OD) of 450 nm. At least three independent experiments were conducted for the biocompatibility assays, and three technical replicates were used in each independent experiment.

### 2.8. In Vivo Mouse Experiment

To establish a mastitis model, the pregnant Kunming mice were used as previously described [16]. In this study, two different strains of *S. aureus* ATCC25923 and *S. aureus* GS1311 were injected via mouse teats to construct infection models of the two different strains, and each group of infection models contained nine mice. The nine mice successfully infected and suffering from mastitis were randomly divided into three groups of three mice each for subsequent experiments: lipopeptide-treated group, antibiotic-treated group and no treatment group. Based on the lipopeptide antimicrobial activity assay results, hemolysis test and cytotoxicity test, C16dKdK was selected for the lipopeptide treatment group. Based on the characteristics of mastitis treatment in local cattle farms and the results of pre-laboratory tests on the drug sensitivity of *S. aureus* ATCC25923 and *S. aureus* GS1311, cefotaxime was selected for the antibiotic treatment group. Mice in the no-treatment group were fed as usual, but the development of mastitis was not treated. Before drug treatment, the mothers were separated from the litter for 2 h. After the mammary glands were filled with milk, the mothers were anesthetized and injected with 10 μg of lipopeptide solution or 25 μg of cefotaxime, respectively, through the mammary ducts. The mothers were caged with their littermates one hour after the drug injection. The drug was administered once a day for 7 days and generally kept for two days after the treatment assay. The status of the mice was observed daily. After the treatment was completed, the mice were euthanized and dissected to observe the mammary tissue lesions. At the same time, the mammary tissues were stripped and sent to Wuhan Xavier Biotechnology Co., Ltd. (Wuhan, China) for tissue sectioning and H&E staining. In contrast, the other mammary gland tissues were used for bacterial load determination and related inflammatory factor expression determination.

### 2.9. Bacterial Load Determination in Breast Tissue

A total of 0.1 g of mammary tissue was weighed from experimental mice, and 0.9 mL of sterile PBS solution was added for tissue homogenization. The tissue homogenate was diluted in a gradient and coated on TSB plates at 100 μL per dilution. The inoculated dishes were incubated overnight at 37 °C in an inverted position. The number of colonies grown on the plates was counted and statistically analyzed.

### 2.10. Inflammatory Factor Expression Assay

Real-time quantitative PCR was used to detect the expression of inflammatory factors in mammary tissues. According to the manufacturer’s instructions, total RNA was extracted from mouse mammary tissue using TRIzol reagent (Invitrogen, Waltham, MA, USA). cDNA was generated by reverse transcription using PrimeScript^TM^RT Master Mix (TAKARA, Kusatsu, Japan) according to the manufacturer’s instructions. The expression of IL-1α, IL-6, IL-10 and TNF-α in breast tissue was measured by qRT-PCR using AceQ^®^ SYBR^®^ Green Master Mix (High ROX Premixed) (Novozymes Biotechnology Co., Ltd., Bagsværd, Denmark) to determine the degree of inflammation. The primer sequences are shown in Table 2. The reaction system was AceQ^®^ qPCR SYBR^®^Green Master Mix 10 μL; PrimerR 0.4 μL; PrimerF 0.4 μL; cDNA, 5 μL; ddH2O, 4.2 μL. The reaction conditions were as follows: pre-denaturation, 95 °C, 5 min; Cycle: 40, 95 °C 10 s, 60 °C 30 s. The results were calculated using the comparative Ct method (2^−ΔΔct^) and normalized to the endogenous level of GAPDH.

### 2.11. Statistical Analysis

At least three independent experiments were conducted for the biocompatibility assays, and three technical replicates were used in each independent experiment. Statistical analysis was performed using a one-way analysis of variance and Student’s *t*-test (two-tailed). The data were analyzed using a statistical package for GraphPad Prism version 8.0 (Chicago, IL, USA). Quantitative data were expressed as mean ± standard deviation, and *p* < 0.01 was considered statistically significant.

## 3. Results

### 3.1. Lipopeptide Design

The molecular structure of the three synthetic lipopeptides is shown in Figure 1. The molecular formula of C16dKdK is C_16_-(D)Lys-(D)Lys-NH_2_, with a theoretical *m*/*z* value of 511.81 and a purity of 94.10%. The molecular formula of C16dKGdK is C_16_-(D)Lys-Gly-(D)Lys-NH_2_, with a theoretical m/z value of 568.86 and a purity of 97.97%. The molecular formula of C16dKGGdK is C_16_-(D)Lys-Gly-Gly-(D)Lys-NH_2_, with a theoretical *m*/*z* value of 625.91 and a purity of 95.56%. The results of the synthesized product spectra were as expected, and the purity of all products was above 90%.

### 3.2. Antibacterial Effect of Synthetic Lipopeptide

By measuring the minimum inhibitory concentration (MIC), we found that all three lipopeptides showed some antibacterial ability against *S. aureus*, with MICs ranging from 0.39–100 μg/mL (Table 3). In addition, we performed scanning electron microscopy on the three lipopeptide-treated bacteria. The results showed that the untreated *S. aureus* was structurally intact, with a smooth surface and entire bacterial cell. 2×MIC C16dKdK-treated *S. aureus* GS1311 could lead to a blurred bacterial structure, cell crumpling and adhesion between the bacteria. 2×MIC C16dKdK-treated *S. aureus* ATCC25923 could lead to apparent cracks on the cell surface of the *S. aureus* bacteria. 2×MIC C16dKGdK and 2×MIC C16dKGGdK, treatment of *S. aureus* GS1311, could lead to different degrees of bacterial cell crumpling. 2×MIC C16dKGdK and 2×MIC C16dKGGdK, treatment of *S. aureus* ATCC25923, could cause roughness and even cracks on the surface of the bacterial cell membrane. The above results indicated that these three lipopeptides could cause different degrees of damage to the bacterial cell structure of *S. aureus* GS1311 and *S. aureus* ATCC25923. Among them, C16dKdK damage is the most obvious (Figure 2).

### 3.3. Hemolytic Analysis

The hemolytic activity of lipopeptides is the most commonly used method for initially assessing the safety of lipopeptide drugs [17]. Hemolytic activity can be one indication of drug safety. The results of the hemolysis test showed that all three lipopeptides had hemolytic activity at high concentrations. When the concentration of C16dKdK is lower than 12.5 μg/mL, more than 85% of red blood cells can remain intact. A total of 6.25 μg/mL C16dKGdK can cause about 20% red blood cell hemolysis. C16dKGGdK showed low hemolytic activity. When the concentration of C16dKGGdK was less than 25 μg/mL, more than 70% of the cells survived (Figure 3B). We used the concentration of lipopeptide that maintains 80% of red blood cell integrity as the maximum concentration for its safe use, so the red blood cell safety concentrations of C16dKdK, C16dKGdK and C16dKGGdK were 12.5 μg/mL, 6.25 μg/mL and 25 μg/mL, respectively.

### 3.4. Cytotoxicity

The assay for cytotoxicity is another critical indicator to verify a drug’s safety. In this study, a CCK8 detection kit was used to detect the cytotoxicity of synthetic lipopeptides on EPH4-Ev cells (Figure 3C). When the concentration of C16dKdK is lower than that of 6.25 μg/mL, more than 70% of the cells can survive. When the concentration of C16dKGdK is lower than that of 3.12 μg/mL, more than 75% of the cells can survive. When the concentration of C16dKGGdK is lower than that of 6.25 μg/mL, more than 75% of the cells can survive. From this, we determined the safe cytotoxic concentrations of C16dKdK, C16dKGdK and C16dKGGdK to be 6.25 μg/mL, 3.12 μg/mL and 6.25 μg/mL, respectively.

### 3.5. Therapeutic Trial

According to the above test results, only C16dKdK has a good bactericidal effect within the safe therapeutic concentration range, so C16dKdK was finally selected as the therapeutic drug for the mouse mastitis model in this study. Since traditional antibiotic treatment is still the most commonly used treatment for mastitis in dairy cows, we combined the selection of drugs commonly used for mastitis treatment in local dairy cows [18] and the results of laboratory drug sensitivity tests on *S. aureus* ATCC25923 and *S. aureus* GS1311 to select cefotaxime as the treatment drug for mice in the antibiotic control group [19]. The results showed that mice in the untreated mastitis group were unresponsive, with a disheveled coat and red and swollen mammary gland area. The mice in the lipopeptide and antibiotic-treated groups were responsive, with a neat coat and no apparent redness and swelling in the mammary glands. The experimental mice were dissected, and pathological changes in mammary gland tissue were observed. The mammary tissue of the mice in the untreated group was congested, and the degree of congestion in the local isolate *S. aureus* GS1311-infected group was more evident than that in the *S. aureus* ATCC25923-infected group. Mammary tissue congestion was reduced in the lipopeptide-treated and antibiotic-treated groups of mice (Figure 4). Notably, the anatomical observation of mice in the two different infection groups presents that the recovery of mammary tissue in the lipopeptide-treated group was better than that in the antibiotic-treated group. The specific performance is that there was no apparent congestion or bleeding in the mammary tissue of mice in the lipopeptide-treated group. In contrast, the mammary tissue of mice in the cefotaxime-treated group was congested and even had pitting bleeding in some areas.

The mammary glands of mice were microscopically observed after tissue sectioning and H&E staining treatment. The results showed a large amount of milk in the mammary ducts of normal mice, and the borders of the mammary ducts were clear. The mice in the untreated mastitis group had significantly reduced or no milk secretion, increased inflammatory cells, such as neutrophils, thickened duct walls, and blurred borders of the nipples. Mice in the lipopeptide-treated and antibiotic-treated groups had increased milk secretion and decreased inflammatory cells, and the borders of the mammary glandular follicles became clear (Figure 5).

To further determine the therapeutic effect of lipopeptide, we measured the bacterial load and the expression of inflammatory factors in the mammary tissue of experimental mice. The results showed that the number of bacteria in the mammary tissue of mice in the lipopeptide-treated group and mice in the antibiotic-treated group was significantly lower than that of mice in the untreated mastitis group. There was no significant difference in the number of bacteria in the mammary tissue of mice in the lipopeptide-treated group and mice in the antibiotic-treated group (Figure 6A,C).

The results of the Inflammatory factor expression assay showed that, in the *S. aureus* GS1311-infected group, the expression levels of IL-1α, IL-6, IL-10 and TNF-α were extremely significantly lower in the mammary tissue of mice in the lipopeptide-treated and antibiotic-treated groups compared with the mammary germs in the untreated mastitis group (Figure 6D). In the *S. aureus* ATCC25923-infected group, the expression level of IL-1α in the mammary tissue of mice in the lipopeptide-treated group was significantly lower than those in the mastitis-untreated group, and the level of IL-1α in antibiotic-treated groups was extremely significantly lower than those in the mastitis-untreated group. In addition, the expression levels of IL-10 in the mammary tissue of mice in the lipopeptide-treated and antibiotic-treated groups were extremely significantly lower than those in the mastitis-untreated group. The expression of TNF-α was significantly lower in mice in the lipopeptide-treated group compared with those in the mastitis-untreated group. The expression of TNF-α was also reduced in the antibiotic-treated group, but the difference was not significant compared with that in the untreated group. The expression of IL-6 was significantly lower in the antibiotic-treated mice compared with the mastitis-untreated group, and the expression of IL-6 was also reduced in the lipopeptide-treated group. However, the difference was insignificant compared to the untreated group (Figure 6B).

## 4. Discussion

Synthetic lipopeptides have recently received attention as substitutes for traditional antibiotics and natural lipopeptides. Attention has primarily been focused on advantages, such as safety, efficiency and reduced drug resistance [9,20]. However, not all synthetic lipopeptides meet the above properties in terms of their practical applications. To determine the lipopeptides that are more compatible with these related properties, we designed and synthesized three antibacterial ultrashort lipopeptides. The corresponding amino acid is dextral (D-amino acid). The standard amino acids in physiology are levorotatory (L-amino acid), while small amounts of D-amino acids also exist in a small amount in biology. Different chiralities in the same amino acid may have different biological functions. The previous research conducted by our laboratory preliminarily found that the antibacterial activity and cytotoxicity of the two lipopeptides, L-C16KGK and D-C16KGK, were different (unpublished data). In this study, the length of the corresponding D-lysine peptide chain is changed by changing the quantity of glycine between the two critical hydrophilic amino acids of the short peptide to find a better corresponding combination.

Lipopeptides are essentially peptides that are coupled to the lipid moiety, which can be made of hydrophilic peptides (cyclic or linear) and hydrophobic fatty acyl chains of amphiphilic character. Fatty acids, as a fundamental component of the membrane, can affect the antibacterial activity and selectivity of antibacterial peptides when they participate in the formation of lipopeptides, and simplicity also extends the half-life of peptides in organisms [11,20,21]. Lipopeptides with from 2 to 4 positive charges and 16 carbon atoms in their lipid chain have been proven to exhibit potent antibacterial activity [15]. Substituting D-amino acids for L-amino acids can minimize cytotoxicity while improving antibacterial activity [22]. Antibacterial peptides containing linear L-amino acids are quickly destroyed by the host protease [10]. Therefore, the fatty acid selected in this study is palmitic acid. All lipopeptides have two positive charges, and all use dextral amino acids to synthesize lipopeptides. The surface of the bacterial cell membrane contains more negative charges. Therefore, it is susceptible to the cationic antimicrobial peptide peptidoglycan phosphopeptide molecules of Gram-positive bacteria that can bind to the cationic antimicrobial peptide by electrostatic adsorption. The interaction with the microbial membrane is the basis of the action mode of cationic antimicrobial peptides, which will destroy the membrane potential, affect the formation of ATP, cause membrane perforation, cause content to flow out, and cause rapid bacterial death [7,23]. Some antimicrobial peptides have been shown to penetrate bacterial cell membranes and may also interfere with other cellular processes, such as DNA replication, transcription and translation to kill bacteria [11,23,24,25]. Our study also showed that morphological changes occurred in the lipopeptide-treated bacterial cells. The C16dKdK-treated group showed a significant collapse of the bacterial body, which could possibly be caused by the lipopeptide disrupting the bacterial cell membrane. However, its specific mechanism of action needs to be further investigated.

The detection of hemolytic activity and cytotoxicity are good indicators when determining the safety of the lipopeptides [26,27]. Therefore, these two indicators were also used in this study to initially assess the safety concentration range of synthetic lipopeptides in experiments to treat mouse mastitis models. Our experimental results determined the red blood cell safety concentrations of C16dKdK, C16dKGdK and C16dKGGdK to be below 12.5 μg/mL, 6.25 μg/mL and 25 μg/mL, respectively. Furthermore, since the antibacterial concentration of lipopeptides should be based on the minimum inhibitory concentration, among these three lipopeptides, only the minimum concentration of C16dKdK was below the cell-safe concentration range. In contrast, the minimum inhibitory concentrations of the other two lipopeptides were much higher than the corresponding safe concentrations. Therefore, C16dKdK was chosen for application in the mouse mastitis treatment experiments. In a study by Arik Makovitzki et al., the MIC of C16KGGK against *S. aureus* was 3.9 μg/mL (6.25 μM), and the hemolytic activity was 8% at 20 μM [6]. We replaced the hydrophilic amino acid K in the lipopeptide molecule with the dK to investigate the effect of different amino acid chirality on the antimicrobial activity of the state of affairs. The antimicrobial activity of C16dKGGdK was diminished. However, when we shortened the peptide chain length, we found that the antibacterial activity of C16dKdK was significantly higher against *S. aureus*, with MICs ranging from 0.39 μg/mL to 1.39 μg/mL. The effect of the change in peptide chain length on its hemolytic activity was insignificant, with 15% hemolytic activity at 20 μM (12.5 μg/mL). Among these three lipopeptides, only C16dKdK has an excellent bactericidal effect in the cell-safe concentration range. This indicates that C16dKdK has more excellent potential for application.

The corresponding pathological models were selected for clinical trials to determine the therapeutic effect of lipopeptides. The glands of cows and mice are functionally and anatomically independent. In addition, each mammary gland has only one teat opening and one dominant duct. Like the cow, this mouse mastitis model provides the unique environment of milk as a pathogenic growth environment and allows for the organism to interact with host cells and immune components, in addition to physical factors, such as lactation [28]. Another fact favorable to mastitis research is that observations of bacterial counts, neutrophil counts, and histological changes in mice are similar to those observed in cows [29,30]. Moreover, mice have the advantage of being small in size, being easy to control, and having a low cost. In this study, a mouse mastitis model was chosen to initially investigate the therapeutic effect of synthetic lipopeptides on mastitis caused by *S. aureus*. We infected *S. aureus* GS1311 and *S. aureus* ATCC25923 by teat injection into the mammary ducts of lactating mice and successfully constructed a mouse mastitis model [31]. At present, there are two diagnostic methods for mastitis, qualitative and quantitative [32]. In the present study, we will combine these two methods to comprehensively determine the severity of mastitis. Our visual assessment of pathological changes and the observation of histopathological sections were qualitative, and statistical changes in bacterial load and the content of inflammatory cytokines were quantitative. Firstly, we observed the ocular pathological changes in the mammary glands of mice in different treatment groups after the treatment experiment. The mammary glands of mice in both lipopeptide-treated and antibiotic-treated groups recovered well, indicating that lipopeptide treatment did not cause further damage to the mammary glands of mice. Even in terms of phenotypic changes, our assessment was that the therapeutic effect of C16dKdK was superior to that of cefotaxime. It has been shown that the C16 peptide can promote blood vessel growth and reduce inflammation in the inflammatory response [33], which may lead to the rapid recovery of mammary gland injury in the C16dKdK-treated group of mice.

Histopathological sections also showed that C16dKdK and cefotaxime have similar therapeutic effects and that the lipopeptides do not cause damage to breast cells. This result proves that C16dKdK is safe for administration via nipple injection. In addition, the degree of bacterial load reduction in the mammary tissue of mice in the lipopeptide treatment group was not significantly different from that in the antibiotic treatment group, indicating that lipopeptides are still able to exert good antibacterial ability in the host cells, and this finding is consistent with the study [8].

When inflammation occurs in the mammary gland, various inflammatory cytokines, including pro-inflammatory and anti-inflammatory cytokines, are involved in the process. The central pro-inflammatory cytokines are TNF-α, IL-1 and IL-6. Among them, TNF-α and IL-1 are essential for mediating the recruitment of leukocytes to the site of the inflammation [34]. TNF-α and IL-6 are immunoreactive in cases of mastitis, while IL-1α of the IL-1 family is more capable of driving the corresponding inflammatory response [35]. These pro-inflammatory cytokines can promote the proliferation of *S. aureus* within the bovine mammary epithelium [36]. In the present study, we evaluated the effect of different treatments on the degree of inflammation in mammary tissue by detecting changes in the expression of TNF-α, IL-1α, and IL-6 in the mammary tissue of mice. The lipopeptide treatment group and antibiotic treatment decreased the expression of these three pro-inflammatory cytokines. In the *S. aureus* ATCC25923-infected group and *S. aureus* GS1311-infected group, the levels of all three pro-inflammatory cytokines, TNF-α, IL-1α, and IL-6, decreased, a result consistent with the treatment effect observed by CHEN Y et al. [37,38], indicating that the inflammatory symptoms were somewhat alleviated. IL-10 is an anti-inflammatory cytokine that is upregulated and involved in the inflammatory process under inflammatory conditions, thus, exerting an immunosuppressive function to reduce the tissue damage caused by excessive and uncontrollable inflammatory effector responses [39]. Both endogenous and exogenous IL-10 strongly inhibit IL-1, IL-6 and TNF-α at the transcriptional level, thus, exerting an anti-inflammatory effect [40]. In our study, IL-10 expression was significantly elevated in mammary cells of mastitis-affected mice, suggesting that IL-10 actively exerts an anti-inflammatory effect at this time to control the extent of the inflammatory response to reduce tissue damage. It has also been shown that the induction of IL-10 facilitates the persistence of bacteria during acute local infections and that a decrease in IL-10 levels and a decrease in various pro-inflammatory cytokines is more conducive to the faster clearance of bacteria by the host [41]. In our study, with the application of therapeutic drugs, the expression of pro-inflammatory cytokines in mouse mammary tissue, mainly TNF-α, IL-1 and IL-6, decreased. At that time, the expression of IL-10, an immunomodulator, also decreased to ensure a gradual return to normal host cells.

## 5. Conclusions

C16dKdK can alleviate the inflammatory reaction of the mammary gland and has a certain therapeutic effect on mouse mastitis, which lays the foundation for the further application of synthetic lipopeptides in clinical treatment. The development of novel antimicrobial lipopeptides could provide an effective therapeutic option to fight infections with drug-resistant bacteria, such as *S. aureus*. This new antibacterial agent could help reduce the use of conventional antibiotics, thereby slowing the development of bacterial resistance to antibiotics and helping to protect human health.

## Figures and Tables

**Figure 1 biomedicines-11-01188-f001:**
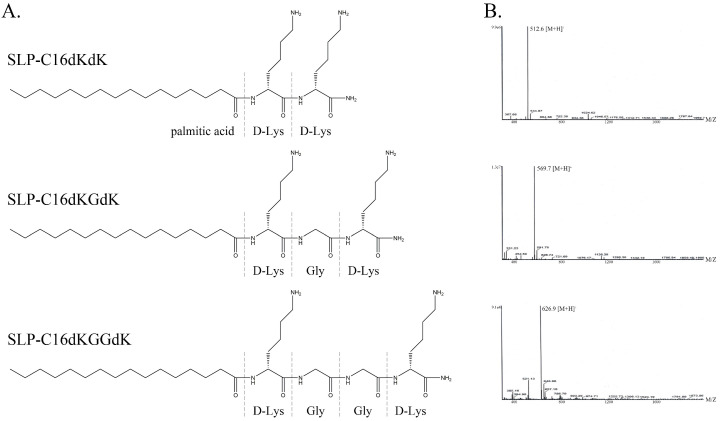
**Lipopeptide synthesis.** (**A**) Synthetic lipopeptides’ (SLPs’) chemical structures. The peptides are represented by the International Union of Pure and Applied Chemistry (IUPAC) single-character symbols. (**B**) Mass spectra of synthetic lipopeptides.

**Figure 2 biomedicines-11-01188-f002:**
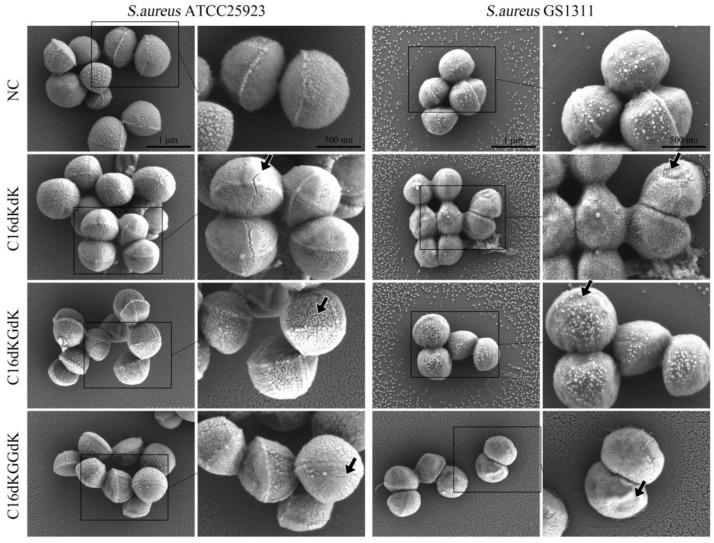
**Lipopeptide antimicrobial activity.** SEM analysis of morphological alterations in *S. aureus* cells after treatment with three lipopeptides. The first column has magnifications of 30 kw× in the various treatment groups, while the second has magnifications of 60 kw×. (The arrows indicate the morphological changes developed on the bacterial surface after treatment with different lipopeptides of *S. aureus*).

**Figure 3 biomedicines-11-01188-f003:**
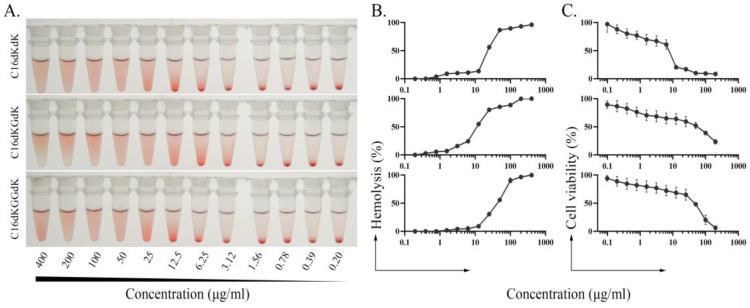
**Hemolytic activity and Cytotoxicity of synthetic lipopeptides.** (**A**) The gradient-diluted lipopeptides were added to a 2% mouse red blood cell PBS suspension and incubated for 1 h at 37 °C. (**B**) 100 μL of the hemolytic activity assay supernatant was pipetted, and the absorbance at OD540 was measured to plot the hemolytic activity function of the lipopeptide. The horizontal coordinate represents the lipopeptide concentration, and the vertical coordinate represents the erythrocyte hemolysis rate. (**C**) CCK8 was used to determine the cytotoxicity of lipopeptides. The horizontal coordinate represents the lipopeptide concentration, and the vertical coordinate represents the cell survival rate.

**Figure 4 biomedicines-11-01188-f004:**
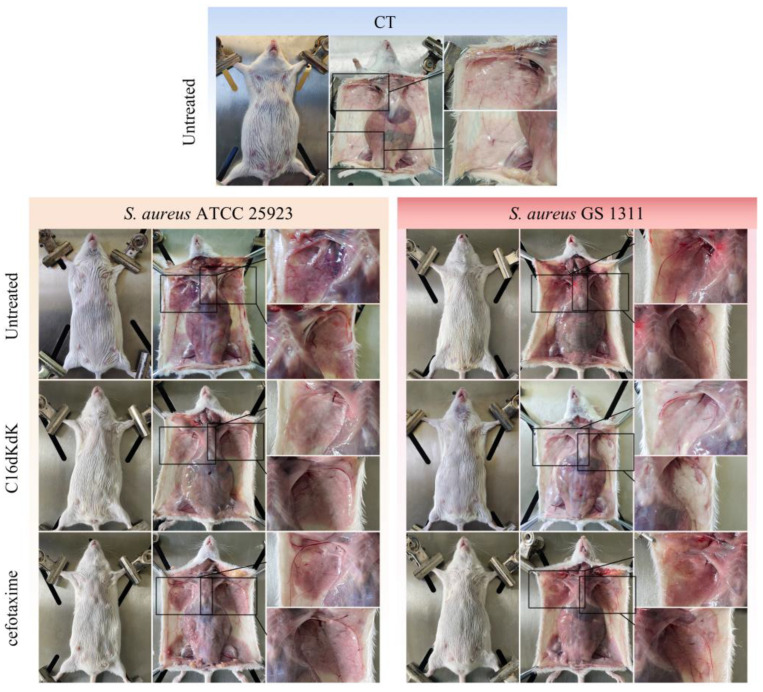
**Mammary gland pathology in mice.** “CT” indicates healthy mice, “ATCC25923” indicates *S. aureus* ATCC25923-infected group, and “GS1311” indicates *S. aureus* GS1311-infected group. “Untreated” indicates that this group of experimental animals did not undergo any treatment; “C16dKdK” indicates that this group of experimental animals received C16dKdK treatment; “cefotaxime” indicates that this group of experimental animals received cefotaxime treatment. The first column of each treatment group shows the mammary appearance of mice, and the second and third columns show the appearance of mammary tissue after the mice were dissected (where the third column shows a magnified image of the corresponding part of the second column, indicated by the rectangles enclosing each area of the images).

**Figure 5 biomedicines-11-01188-f005:**
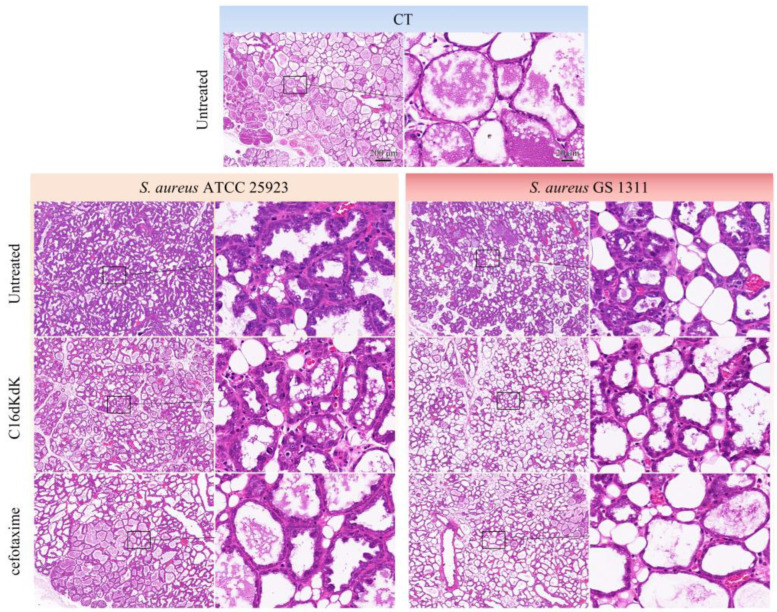
**Histopathological examination of mammary glands of mice.** The mammary gland tissues were fixed in formalin and then sectioned and stained with H&E. Microscopic observation. “CT”, “ATCC25923,” and “GS1311” denote the healthy mice group, *S. aureus* ATCC25923-infected group and *S. aureus* GS1311-infected group, respectively. “Untreated” indicates no treatment, “C16dKdK” indicates treatment with C16dKdK after infection, and “cefotaxime” indicates treatment with cefotaxime after infection. Each group’s left and right columns are shown in the images collected at 10× and 40×, respectively.

**Figure 6 biomedicines-11-01188-f006:**
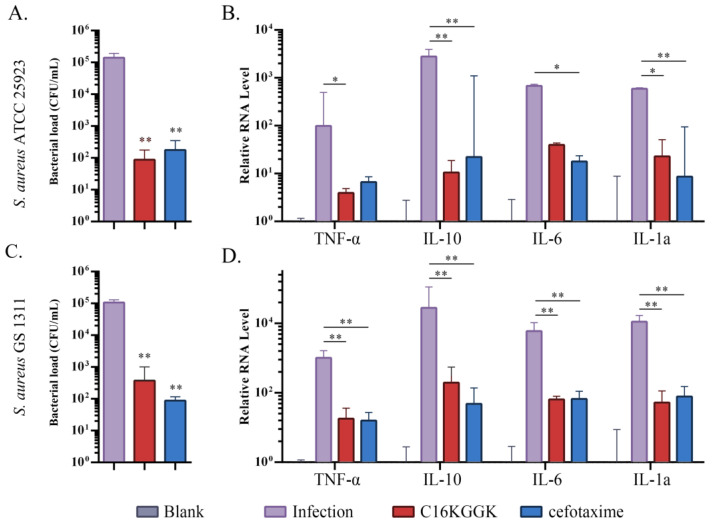
**Quantitative determination of changes in mammary gland tissue of the mice.** (**A,C**) Intra-tissue bactericidal effect of SLPs on mastitis caused by *S. aureus*. (**B**,**D**) Expression of inflammatory factors in mammary tissue of mice before and after SLPs treatment (“*”: *p* < 0.01; “**”: *p* < 0.001).

**Table 1 biomedicines-11-01188-t001:** The sequence of the synthetic lipopeptide.

Name	Sequence
C16dKdK	C_16_-(D)Lys-(D)Lys-NH_2_
C16dKGdK	C_16_-(D)Lys-Gly-(D)Lys-NH_2_
C16dKGGdK	C_16_-(D)Lys-Gly-Gly-(D)Lys-NH_2_

**Table 2 biomedicines-11-01188-t002:** The primer sequences of real-time quantitative PCR.

Inflammatory Factors	Primer	Primer Sequence
IL-1α	m * -il1α-qF	ACCCAGATCAGCACCTTACACC
	m-il1α-qR	CTCCTCCCGACGAGTAGGCAT
IL-6	m-il6-qF	TCTGGAGCCCACCAAGAACGA
	m-il6-qR	ACATGTGTAATTAAGCCTCCGAC
IL-10	m-il10-qF	CTGCACCCACTTCCCAGTCG
	m-il10-qR	ACTGGATCATTTCCGATAAGGC
TNF-α	m-tnf-qF	AGCACAGAAAGCATGATCCG
	m-tnf-qR	AGCTGCTCCTCCACTTGGT
GAPDH	m-gapdh-qF	GCGACTTCAACAGCAACTCCC
	m-gapdh-qR	CACCCTGTTGCTGTAGCCGTA

* m means mouse.

**Table 3 biomedicines-11-01188-t003:** MIC of the synthetic lipopeptide.

MIC (μg/mL)	C16dKdK	C16dKGdK	C16dKGGdK
ATCC25923	0.39	25	12.5
GS1311	1.56	50	100

## Data Availability

Not applicable.

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
