# Peer review of "Synthetic Cationic Lipopeptide Can Effectively Treat Mouse Mastitis Caused by Staphylococcus aureus"

_biomedicines, 2023, doi:10.3390/biomedicines11041188_

Round 1

Reviewer 1 Report

The paper of  Peng Jie, which deals with testing new drugs lipopeptides' antibacterial activity against Staphylococcus aureus, responsible for mastitis in cattle, I think is very interesting. The only thing that puzzles me is the use of a mouse model for the study of mastitis. Although the mouse is often used as a mastitis model, I agree with the conclusions of the work of 2015 by Wendy V Ingman, Danielle J Glynn & Mark R Hutchinso, Mouse models of mastitis - how physiological are they?.

Beyond the usefulness of the animal model, there are minor revisions that must necessarily be made.

-In the section of materials and methods, the final part of paragraph 2.8, there is a reference to an unspecified 'quantitative fluorescence PCR'. I think the authors want to indicate an RT- Real Time PCR. It is suggested to create a separate paragraph in which all the details of the technique are specified, as well as the sequences of the primers or any probes used.

-In Graph C of Figure 3 the error bands are very large and strangely missing for some points. It is advisable to redo the experiment and then the graph, inserting the error bands in all the samples.

-Figure 4 must be split so that it can be fattened and therefore easier to visualize. In addition, it is recommended to improve the contrast of immunohistochemistry always to better appreciate the anibacterial activity of the various compounds.

-Figure 5 is completely missing.

Author Response

Response to Reviewer 1 Comments

Point 1: The paper of  Peng Jie, which deals with testing new drugs lipopeptides' antibacterial activity against Staphylococcus aureus, responsible for mastitis in cattle, I think is very interesting. The only thing that puzzles me is the use of a mouse model for the study of mastitis. Although the mouse is often used as a mastitis model, I agree with the conclusions of the work of 2015 by Wendy V Ingman, Danielle J Glynn & Mark R Hutchinso, Mouse models of mastitis - how physiological are they?

Response 1: Thank you for your question, and we apologize for not addressing this issue in the article. The mouse is still considered a very suitable tool for research focusing on the pathogenesis and control of bovine intramammary infections. Both species have glands that are functionally and anatomically independent from each other. In addition, each mammary gland has only one teat opening and one primary duct. As in cows, this mouse mastitis model also provides milk's unique pathogen growth environment and allows interaction of the organism with the host cells and immune components, besides offering physical factors such as suckling. Also beneficial for the study of mastitis is that observations in mice concerning bacterial counts, neutrophil numbers and histological changes are similar to those in cows. Based on the above findings, this study was selected to initially investigate the therapeutic effect of synthetic lipopeptides on mastitis caused by Staphylococcus aureus in a mouse mastitis model. We have added this section to the revised manuscript, specifically in Discussion, page 11, lines 471-478.{[ref.] Notebaert, S.; Meyer, E., Mouse models to study the pathogenesis and control of bovine mastitis. A review. VET QUART 2006, 28, 2-13.}.

Point 2: In the section of materials and methods, the final part of paragraph 2.8, there is a reference to an unspecified 'quantitative fluorescence PCR'. I think the authors want to indicate an RT- Real Time PCR. It is suggested to create a separate paragraph in which all the details of the technique are specified, as well as the sequences of the primers or any probes used.

Response 2: Thank you for your valuable comments. In the newly submitted manuscript, we have rewritten the material methods section and added all the details of the Real-Time PCR experiments, specifically in Materials and Methods, page 4, lines 168-180.

Point 3: In Graph C of Figure 3 the error bands are very large and strangely missing for some points. It is advisable to redo the experiment and then the graph, inserting the error bands in all the samples.

Response 3: Thank you for your valuable comments. We have redone the experiment and statistical analysis of the experimental results. The new experimental results have been submitted in the manuscript. Specifically, in the results, p. 7, lines 254-262.

Point 4: Figure 4 must be split so that it can be fattened and therefore easier to visualize. In addition, it is recommended to improve the contrast of immunohistochemistry always to better appreciate the anibacterial activity of the various compounds.

Response 4: Thank you for your valuable comments. In the newly submitted manuscript, we have split this image into two completely new images (Figure 4 and Figure 5 in the new manuscript). In addition, we adjusted the contrast of the immunohistochemistry results to make the revised images easier to understand. Specifically in Results, page 8, lines 290-313 (Figure 4); page 9, lines 331-352 (Figure 5).

Point 5: Figure 5 is completely missing.

Response 5: We appreciate the reviewer's reminder. We have re-uploaded this image in the new manuscript, and it is marked as Figure 6.Specifically in Results, page 10, lines 386-402.

Reviewer 2 Report

Peng et al.:  Synthetic cationic lipopeptide can effectively treat mastitis caused by Staphylococcus aureus

Mastitis due to Staphylococcus aureus in dairy cows is common and costly.   The authors note that the disease is increasingly difficult to treat because the widespread use of traditional antibiotics has led to drug-resistant strains of bacteria.  Recently, according to the authors, lipopeptide antibiotics have become increasingly important to address this gap in effective antibiotics.  The authors designed and synthesized three cationic lipopeptides with palmitic acid, all of which contained two positive charges and all with dextral amino acids.  Scanning electron microscopy and MIC determinations were used to evaluate activity against S. aureus.  The safety concentration was assessed using mouse erythrocyte susceptibility and CCK8 cytotoxicity tests.  Then, a lipopeptide with bacterial activity and appropriate safety concentrations were tested in a mouse mastitis model.  The authors concluded that one of the three candidates with high activity was found to be able to treat S. aureus mastitis, as tested using the mouse model.  Finally, the authors conclude that their research can serve as a starting point for research into effective antibiotics for mastitis in dairy cows. 

Research investigating novel antibiotics for treatment of bovine mastitis is important in order to provide alternatives to traditional antibiotics.  This manuscript addresses and presents new information on this important topic.  This reviewer has concerns with the manuscript in three areas, including: Overstatement of the significance of the results in some areas, writing that is in the wrong tense, and inadequate description of methods.    These concerns are detailed among specific comments for the authors to consider:

Title, p. 1, ll. 2-3:  The title states that the lipopeptide “..can effectively treat mastitis…”  In my opinion, this over-states the findings from the research and should be more conservatively stated.  If you wish to keep this portion of the title, then I suggest that you modify it in some way to indicate that it is effective in “mouse mastitis” or “in a mouse model for mastitis” or some similar change.  In my opinion, your findings in the mouse model do not provide evidence to state that the lipopeptides can effectively treat mastitis, for example, in dairy cattle.

Abstract, p. 1, ll. 15-17:  Is the “…used in lipopeptide synthesis in this study” at the end of this sentence necessary?  Could it be deleted?

Abstract, p. 1, ll. 19-20:  Here you refer to the “safety concentration range of lipopeptides for clinical usage was then determined..”  I will make specific comments about this under another section, but I think you should more conservatively state this, but by saying that the ‘safety concentrations were ESTIMATED…”

Abstract, p. 1, l. 22:  Can you state specific bacteria or limit the statement to S. aureus?

Introduction, p. 1, l. 30:  Here, and after, please add a space between the S. and the aureus.  S. aureus, not S.aureus.

Introduction, p. 1, l. 33:  “… at the breast.”   When used in reference to the bovine, it is more correctly referred to as “mammary gland.”

Introduction, p. 2, ll. 49-50:  Revision of this sentence would make it easier to understand—something like this:  “Creation of a novel class of short cationic lipopeptides involves connecting dipeptides, tripeptides and tetrapeptides to palmitoyl chains via their N-terminus.”

Introduction, p. 2, ll. 57-58:  Was there a particular reason why palmitic acid was chosen as the fatty acid?

Introduction, p. 2, last 2 lines, ll. 59-60:  Here you state:  “The results demonstrated that the associated lipoprotein effectively treated S. aureus-caused cow mastitis.”  It does not seem appropriate to give results here.   It would be appropriate to give the purpose(s) of the study or hypotheses here.

Materials & Methods (M&M), p. 2, ll. 65-67:  Four comments in this section:  1.  I didn’t look it up, but is Gansu a complete location description?   2.  It would be useful to indicate what kind of mastitis—clinical or subclinical?   3.  Last sentence:   It is written as “bacterial”—assume you meant to say “bacteria”?   4.  How long did you incubate the inoculated media?  Overnight?  Number of hours?     

M&M, p. 2, ll. 79-81:  Correct the way this is written:  “ synthesized BY Suzhou…” OR “….OBTAINED from..”

M&M, p. 2, ll. 82-84:  You state:  “The purity and structure of the three have reached the expected goal according to the analysis of high performance liquid chromatography and mass spectrometry.”   This statement is not enough for the reader to judge the information.   Can you give more detail?

M&M, p. 2, l. 87:  This sentence can be re-stated as:  “The dilution method WAS USED TO determine the minimum inhibitory concentration…”

M&M, p. 3, l. 90 and l. 96—and after:  In this section, many of the statements read as if they are instructions for someone to follow, rather than a description of what was done.  On l. 90, you state “Diluted the prepared lipopeptide…”   This would be better written as “The prepared lipopeptide solution was diluted…”  On l. 96, you state:  “Put the strain in TSB culture medium and cultivated…”  This would be better written as “The bacterial strain was PLACED (or similar verb) in TSB culture medium…”  There are several additional sentences like l. 90 and l. 96.  I did not comment on those individually and will leave it to the authors to identify and modify those.

M&M, p. 3, l. 95:  “Assay” is incorrectly written as “assey.”

M&M, p. 3, ll. 95-110:  Here and in some other sections, insufficient details are given on methods:  For instance, on l. 102, you note rinsing in phosphate buffer, but give no indication of approximate volume.  On l. 107, you mention drying but give no indication of temperature. 

M&M, p. 3, ll. 98-99:  It would be good if you would give the specific “electron microscope fixative.”  What do you mean by “blow open” on l. 99?  And “blew” on l. 101.

M&M, p. 3, ll. 99-100:  This sentence is awkward and not clear:  “One group did not do any treatment to be observe the normal strain status as a negative control.”  Please re-write for clarity.

M&M, p. 3, ll. 111-122:  Same comment here on details.  Another way to deal with some of this would be to say something like:  “The hemolytic assay was done (exactly) as described by Lvfeng, with the exception that “…………..”

M&M, p. 3, ll. 131-132:  You state that:  “The mouse mastitis model was constructed by injecting bacteria directly into the mammary tissue of mice.”  You give references 10 and 11 for this.  The references give somewhat different approaches to this procedure.  Again, if correct and possible, you could state that the procedure was done exactly as described by Chandler (or the other reference), with the following exceptions….. 

Results, p. 4, ll. 164-165:  What does SLP stand for?  What does IUPAC stand for?

Results, p. 4, ll. 167-175:  It may be my shortcoming, but I don’t clearly see the changes described.  Perhaps you could use arrows or some other means to point out the changes?

Results, p. 5, l. 187:  Here you state that:  “The detection of hemolytic activity can well verify drug safety.”  This statement appears to be a gross under-estimation of what is required to assure drug safety is determined.  It may not actually be correct.  If you have a reference or documentation that hemolytic activity is a direct indicator of drug safety—if that is the case.  I believe that hemolytic activity can be a part of assessing drug safety, but certainly not a sole/total indicator.

M&M, p. 6, l. 210:  Here you state that “Following centrifugation, a PIECE of the supernatant was aspirated……”  Not sure what you mean by “PIECE” here?

M&M, p. 6, l. 220:  Please explain why you chose a cephalosporin and, specifically, cefotaxime?

M&M, In several instances (some examples follow) you make statements about judgements and comparisons and that seem to suggest that statistical comparisons were done.  Please give more details or remove the indications of statistical comparisons:

  P. 6, ll. 225-226:  “…..were relatively more mentally active than those in the untreated group.”

  P. 6, l. 227:  “….without significant improvement…”

  P. 7, l. 247:  “…bacterial load was significantly lower….”

Results, p. 7, ll. 257-258:  Two sentences here seem incomplete—or, at least, not clear to me:  “The treatment means therapeutic drugs for post-infection mice.  Untreated means no applicable.” 

Results, p. 7, ll. 263-265:  A Figure 5 is referenced here, but I see no Figure 5.

Discussion, p. 7, ll. 267-269:  This first sentence is long and complex—should be simplified.  One SUGGESTION:  Synthetic lipopeptides have recently received attention as substitutes for traditional antibiotics and natural lipopeptides.  The attention has primarily focus ed on advantages such as safety, efficiency and reduced drug resistance.

Discussion, p. 8, ll. 295-298:  You should provide a reference for this sentence.

Discussion, p. 8, ll. 302-303:  Again, as remarked previously, hemolytic activity and cytotoxicity provide good information about drug effects, but certainly are not stand-alone measures of drug safety.

Discussion, p. 9, ll. 347 and l. 350 and l. 376:  Breast tissue is used in each of these situations and mammary gland would be preferred.

References: There is inconsistent use of bold vs. non-bold in titles.

Author Response

Response to Reviewer 2 Comments

Research investigating novel antibiotics for treatment of bovine mastitis is important in order to provide alternatives to traditional antibiotics. This manuscript addresses and presents new information on this important topic. This reviewer has concerns with the manuscript in three areas, including: Overstatement of the significance of the results in some areas, writing that is in the wrong tense, and inadequate description of methods. These concerns are detailed among specific comments for the authors to consider:

We sincerely thank the reviewers for their careful reading. Based on the reviewers' suggestions, we have revised the description of the role of lipopeptides in the article in order to convey their importance more accurately. We have also revised the material and methods section of the article to be more detailed. Finally, we have submitted the revised manuscript to the MDPI language retouching system for revision. We hope that you will accept the final manuscript.

Point 1: Title, p. 1, ll. 2-3: The title states that the lipopeptide “..can effectively treat mastitis…” In my opinion, this over-states the findings from the research and should be more conservatively stated. If you wish to keep this portion of the title, then I suggest that you modify it in some way to indicate that it is effective in “mouse mastitis” or “in a mouse model for mastitis” or some similar change. In my opinion, your findings in the mouse model do not provide evidence to state that the lipopeptides can effectively treat mastitis, for example, in dairy cattle.

Response 1: We apologize for our lack of rigor. We have corrected the title to "Synthetic cationic lipopeptide can effectively treat mouse mastitis caused by Staphylococcus aureus" based on your suggestion. Thank you for your correction. The change's specific location is in the article's title section, page 1, lines 2-3.

Point 2: Abstract, p. 1, ll. 15-17:  Is the “…used in lipopeptide synthesis in this study” at the end of this sentence necessary? Could it be deleted?

Response 2: We appreciate your suggestion and have now removed "...used in lipopeptide synthesis in this study". The specific change is in abstract page 1, lines 16-17.

Point 3: Abstract, p. 1, ll. 19-20: Here you refer to the “safety concentration range of lipopeptides for clinical usage was then determined..” I will make specific comments about this under another section, but I think you should more conservatively state this, but by saying that the ‘safety concentrations were ESTIMATED…”

Response 3: We appreciate your correction and we have now corrected "determined" to "estimated". The exact location of the change is in the abstract section, page 1, line 19.

Point 4: Abstract, p. 1, l. 22: Can you state specific bacteria or limit the statement to S. aureus?

Response 4: We apologize for the lack of rigor. The "bacteria" in this sentence limits the statement to S. aureus, and we have corrected the "bacteria" in the manuscript to We have corrected "bacteria" in the manuscript to "S. aureus" The change is in the abstract, page 1, line 24.

Point 5: Introduction, p. 1, l. 30: Here, and after, please add a space between the S. and the aureus. S. aureus, not S. aureus.

Response 5: We apologize for our carelessness. In the resubmitted manuscript, we have added spaces between S. and aureus in all places in the text where S. aureus is involved. Thank you for your correction! Revisions are in the full text, lines 11, 24, 25, 31, 36, 37, 39, 40, 67, 70, 108, 213, 215, 219, 315, 404, and 481.

Point 6: Introduction, p. 1, l. 33: “… at the breast.” When used in reference to the bovine, it is more correctly referred to as “mammary gland.”

Response 6: We appreciate your correction. We have now corrected "breast" to "mammary gland". Revisions is in abstract, page 1, line 34.

Point 7: Introduction, p. 2, ll. 49-50: Revision of this sentence would make it easier to understand—something like this: “Creation of a novel class of short cationic lipopeptides involves connecting dipeptides, tripeptides and tetrapeptides to palmitoyl chains via their N-terminus.”

Response 7: We appreciate your correction. In the new manuscript, we have revised "To create a novel class of short cationic lipopeptides, dipeptides, tripeptides, and tetrapeptides are connected to palmitoyl chains via their N-terminus" to "The creation of a novel class of short cationic lipopeptides involves connecting dipeptides, tripeptides and tetrapeptides to palmitoyl chains via their N-terminus" as you suggested. The revision is in the Introduction, page 2, lines 52-54.

Point 8: Introduction, p. 2, ll. 57-58: Was there a particular reason why palmitic acid was chosen as the fatty acid?

Response 8: We thank you for your helpful questions. We chose palmitic acid because the case of chemical surfactants shows that the longer the lipophilic acyl chain, the more influential the micellization due to an increase in intermolecular hydrophobic interactions. It was found that using palmitic acid in lipopeptides resulted in good bactericidal activity of the lipopeptides. We have described this issue more clearly in the revised manuscript. The revised position is in the Introduction, page 2, lines 57-61

Point 9: Introduction, p. 2, last 2 lines, ll. 59-60: Here you state: “The results demonstrated that the associated lipoprotein effectively treated S. aureus-caused cow mastitis.” It does not seem appropriate to give results here. It would be appropriate to give the purpose(s) of the study or hypotheses here.

Response 9: We apologize for the laxity of our language description. In the new manuscript, we have changed this section to “The aim of this study was to explore whether the associated lipopeptide can effectively treat S. aureus-caused mouse mastitis.” The revision is in the Introduction, page 2, lines 66-67.

Point 10: Materials & Methods (M&M), p. 2, ll. 65-67: Four comments in this section: 1. I didn’t look it up, but is Gansu a complete location description? 2. It would be useful to indicate what kind of mastitis—clinical or subclinical? 3. Last sentence: It is written as “bacterial”—assume you meant to say “bacteria”? 4. How long did you incubate the inoculated media? Overnight? Number of hours?

Response 10: We thank you for your careful reading and apologize for the unclear description. Gansu is a province located northwest of China, and Gansu Agricultural University is an agricultural university in Gansu, China. The Staphylococcus aureus strain GS1311 used in this paper was isolated from a local cow's milk with clinical-type mastitis (revised in line 73). In this paragraph, we apologize for our carelessness, the description of "bacterial" was indeed wrong, and we corrected it to "bacterial passaging cultures" (revised in line 74). In addition, we have overnight incubated the bacteria in the medium for preservation and passaging (revised in line 75). Finally, we have revised and clarified the above suggestions made by the reviewers in the newly submitted manuscript. Thank you for your corrections.

Point 11: M&M, p. 2, ll. 79-81: Correct the way this is written: “synthesized BY Suzhou…” OR “….OBTAINED from..”

Response 11: We thank you for your suggestion. In the revised manuscript, we have corrected "synthesized from Suzhou..." to "synthesized by Suzhou...". The revision is in Materials and Methods, page 2, line 88.

Point 12: M&M, p. 2, ll. 82-84: You state: “The purity and structure of the three have reached the expected goal according to the analysis of high performance liquid chromatography and mass spectrometry.” This statement is not enough for the reader to judge the information. Can you give more detail?

Response 12: We apologize for our carelessness. Since Figure 1, attached to the first manuscript submission, was missing the results of the mass spectrometric assay of the synthesized lipopeptides, in our resubmitted manuscript, we have added this section and submitted the synthesis reports of the three lipopeptides in the addendum to the article. We hope the newly uploaded results will answer your questions and thank you very much for your corrections! Revision in Results, page 5, lines 206-209.

Point 13: M&M, p. 2, l. 87: This sentence can be re-stated as: “The dilution method WAS USED TO determine the minimum inhibitory concentration…”

Response 13: We appreciate the valuable comments of the reviewers. We have now revised the "The dilution method determined the minimum inhibitory concentration …" in the original manuscript to "The minimum inhibitory concentrations (MICs) of three peptides were measured by the microbroth dilution method…" in order to be able to express our views more clearly. The revision is in Materials and Methods, p. 3, lines 97-99.

Point 14: M&M, p. 3, l. 90 and l. 96—and after: In this section, many of the statements read as if they are instructions for someone to follow, rather than a description of what was done. On l. 90, you state “Diluted the prepared lipopeptide…” This would be better written as “The prepared lipopeptide solution was diluted…” On l. 96, you state: “Put the strain in TSB culture medium and cultivated…” This would be better written as “The bacterial strain was PLACED (or similar verb) in TSB culture medium…” There are several additional sentences like l. 90 and l. 96. I did not comment on those individually and will leave it to the authors to identify and modify those.

Response 14: We are very grateful to you for your valuable comments. In the revised manuscript, we have rewritten all relevant paragraphs to express our views more clearly, and we have revised any unreasonable descriptions of the materials and methods. We have now corrected "Diluted the prepared lipopeptide..." to "The peptide solutions were diluted twofold with the final concentrations of..." (revision in Materials and Methods, p. 3, lines 100-101); have corrected "Put the strain in TSB culture medium and cultivated..." has been corrected to "In brief, the mid-log phase S. aureus ATCC25923 and GS1311 cells (1× 108 CFU/mL) were incubated with..." (revision in Materials and Methods, p. 3, lines 108-110).

Point 15: M&M, p. 3, l. 95: “Assay” is incorrectly written as “assey”.

Response 15: We apologize for our carelessness. We have corrected the typo in our new manuscript submission and appreciate your correction! Revision in Materials and Methods, page 3, line 117.

Point 16: M&M, p. 3, ll. 95-110: Here and in some other sections, insufficient details are given on methods: For instance, on l. 102, you note rinsing in phosphate buffer, but give no indication of approximate volume. On l. 107, you mention drying but give no indication of temperature.

Response 16: We appreciate your valuable comments and apologize for the lack of clarity in the description of the material methods section. We have rewritten this section in the newly submitted manuscript based on your suggestions. The revision is in Materials and Methods, page 4, lines 105-116.

Point 17: M&M, p. 3, ll. 98-99: It would be good if you would give the specific “electron microscope fixative.” What do you mean by “blow open” on l. 99? And “blew” on l. 101.

Response 17: We appreciate your valuable comments. The electron microscope sample fixation was done using 2.5% glutaraldehyde. We apologize for the incorrect description of this part and have rewritten it in the newly submitted manuscript. The revision is in Materials and Methods, page 4, lines 105-116.

Point 18: M&M, p. 3, ll. 99-100: This sentence is awkward and not clear: “One group did not do any treatment to be observe the normal strain status as a negative control.” Please re-write for clarity.

Response 18: Thank you for your valuable comments. We have rewritten the paragraph. We hope that the revised description will accurately express the article's content. Revised in Materials and Methods, page 4, lines 105-116.

Point 19: M&M, p. 3, ll. 111-122: Same comment here on details. Another way to deal with some of this would be to say something like: “The hemolytic assay was done (exactly) as described by Lvfeng, with the exception that “…………..”

Response 19: Thank you for your valuable comments. In the newly submitted manuscript, we have rewritten this section and hope that the revised content will be more concise and clear. Revision in Materials and Methods, p. 3, lines 118-127.

Point 20: M&M, p. 3, ll. 131-132: You state that: “The mouse mastitis model was constructed by injecting bacteria directly into the mammary tissue of mice.” You give references 10 and 11 for this. The references give somewhat different approaches to this procedure. Again, if correct and possible, you could state that the procedure was done exactly as described by Chandler (or the other reference), with the following exceptions…..

Response 20: Thank you for your valuable comments. The method used in our study was to inject bacteria directly into mouse mammary tissue, a method consistent with that in reference 10 (reference 14 in the new manuscript) and slightly different from reference 11, where several methods are provided, including the one using a syringe that requires clipping of the mouse teat, which we believe is prone to other pathological damage and therefore The nipples were not clipped. In the newly submitted manuscript, we have removed reference 11 and described this section more clearly. Revision in Materials and Methods, p. 4, lines 139-140.

Point 21: Results, p. 4, ll. 164-165: What does SLP stand for? What does IUPAC stand for?

Response 21: Thank you for your valuable comments. We apologize for the lack of clarity in our previous manuscript submission regarding the abbreviations SLP and IUPAC. SLP stands for "Synthetic lipopeptide" and IUPAC stands for "International Union of Pure and Applied Chemistry." In the newly submitted manuscript, we further explained these two words. Revision in Results, page 5, lines 207-209.

Point 22: Results, p. 4, ll. 167-175: It may be my shortcoming, but I don’t clearly see the changes described. Perhaps you could use arrows or some other means to point out the changes?

Response 22: We appreciate your valuable comments. We apologize for the lack of detail in our labeling of the images. In the newly submitted manuscript, the changes that occurred in the experimental group have been indicated with arrows. Revision in Results, page 6, line 227.

Point 23: Results, p. 5, l. 187: Here you state that: “The detection of hemolytic activity can well verify drug safety.” This statement appears to be a gross under-estimation of what is required to assure drug safety is determined. It may not actually be correct. If you have a reference or documentation that hemolytic activity is a direct indicator of drug safety—if that is the case. I believe that hemolytic activity can be a part of assessing drug safety, but certainly not a sole/total indicator.

Response 23: We are apology that the statement "The detection of hemolytic activity can well verify drug safety." exaggerate the role of hemolysis tests in drug safety assessment. The antimicrobial effect of lipopeptides depends mainly on their lipophilic activity. Cell membranes also have many lipid components, which leads to the fact that most lipopeptides have high erythrocyte hemolytic activity along with their bactericidal effect, so the detection of hemolytic activity of lipopeptides is the most commonly used method to determine the safety assessment of lipopeptide drugs initially. We apologize for the inaccurate description in the manuscript, and we have rewritten this section in the newly submitted manuscript. This will provide a more accurate representation of the role of the hemolysis test in drug safety assessment. Revision in Results, page 6, lines 232-233.

Point 24: M&M, p. 6, l. 210: Here you state that “Following centrifugation, a PIECE of the supernatant was aspirated……” Not sure what you mean by “PIECE” here?

Response 24: We appreciate your close inspection and apologize for the inaccuracy of our description. We have removed "a piece of" in the newly uploaded manuscript. We hope that the revised description will better express our research. Revision in Results, page 7, lines 255-257.

Point 25: M&M, p. 6, l. 220: Please explain why you chose a cephalosporin and, specifically, cefotaxime?

Response 25: Thank you very much for your valuable comments. Due to an oversight on our part, it was not clearly stated in the original manuscript. Cephalosporins are commonly used locally for the treatment of mastitis in dairy cows. Furthermore, in our previous study, we tested the drug sensitivity of local isolates of Staphylococcus aureus (including GS1311 involved in this study), causing mastitis in dairy cows. The results showed that GS1311 was sensitive to cefotaxime (results not published). Therefore, we chose cefotaxime as a control group for antibiotic treatment to compare the effectiveness of conventional antibiotics and synthetic lipopeptides in mastitis treatment. This section and related references have been added to our newly submitted manuscript. Modified in Results, p. 7, lines 266-271.

Point 26: M&M, In several instances (some examples follow) you make statements about judgements and comparisons and that seem to suggest that statistical comparisons were done. Please give more details or remove the indications of statistical comparisons:

P. 6, ll. 225-226: “…..were relatively more mentally active than those in the untreated group.”

P. 6, l. 227:“….without significant improvement…”

P. 7, l. 247: “…bacterial load was significantly lower….”

Response 26: We appreciate your valuable comments, and in the new manuscript, we have revised "...were relatively more mentally active than those in the untreated group" to "The results showed that mice in the untreated mastitis group were unresponsive ..."; delete "…without significant improvement…"; revise "…bacterial load was significantly lower…" to "The results showed that the number of bacteria in the mastitis group was unresponsive that the number of bacteria in the mammary tissue of mice...". Furthermore, we revised all the relevant paragraphs, hoping to express the article's ideas more clearly. The changes were made in Results, page 7, lines 271-274; page 8, lines 326-330.

Point 27: Results, p. 7, ll. 257-258: Two sentences here seem incomplete—or, at least, not clear to me: “The treatment means therapeutic drugs for post-infection mice. Untreated means no applicable.”

Response 27: Thank you for your valuable comments. We apologize for the unclear descriptions in the manuscript, and in the new manuscript, we have split the original Figure 4 (Figures 4 and 5) and written new figure notes. We hope that the revision will clearly express our views. Thank you for your corrections! Revisions in Results, p. 8, lines 290-322 (Figure 4); p. 9, lines 331-359 (Figure 5).

Point 28: Results, p. 7, ll. 263-265: A Figure 5 is referenced here, but I see no Figure 5.

Response 28: We appreciate the reviewer's reminder. We have re-uploaded this image in the new manuscript, and it is marked as Figure 6. Results, page 10, lines 386-402.

Point 29: Discussion, p. 7, ll. 267-269: This first sentence is long and complex—should be simplified.  One SUGGESTION: Synthetic lipopeptides have recently received attention as substitutes for traditional antibiotics and natural lipopeptides. The attention has primarily focused on advantages such as safety, efficiency and reduced drug resistance.

Response 29: We are very grateful for your valuable comments. We have revised "In recent years, synthetic lipopeptide, as a substitute for traditional antimicrobial peptides and natural lipopeptides, has attracted extensive attention due to its advantages, such as safety, efficiency, and difficulty in drug resistance" to "Synthetic lipopeptides have recently received attention as substitutes for traditional antibiotics and natural lipopeptides. Attention has primarily been focused on advantages such as safety, efficiency and reduced drug resistance." in the revised manuscript. Revision in Discussion, p. 10, lines 408-410.

Point 30: Discussion, p. 8, ll. 295-298: You should provide a reference for this sentence.

Response 30: We are very grateful for your comments. In the newly submitted manuscript, we have inserted references for this section, i.e., references 5 and 21 in the revised manuscript. Revision in Discussion, p. 11, line 441.

Point 31: Discussion, p. 8, ll. 302-303: Again, as remarked previously, hemolytic activity and cytotoxicity provide good information about drug effects, but certainly are not stand-alone measures of drug safety.

Response 31: Thank you very much for your valuable comments. We apologize that our description in the original manuscript was inaccurate, and we have rewritten the relevant parts of the new manuscript. Both the hemolysis and cytotoxicity tests are used as indicators to assess the drug's safety, respectively. Both can be used in this study to assess the safety range of the lipopeptide drug under study. Revised in Discussion, p. 11, lines 448-451.

Point 32: Discussion, p. 9, ll. 347 and l. 350 and l. 376: Breast tissue is used in each of these situations and mammary gland would be preferred.

Response 32: We are very grateful to your valuable comments and we have revised "Breast tissue" to "mammary gland". Revision in Discussion, page 12, lines 497, 501, 533.

Point 33: References: There is inconsistent use of bold vs. non-bold in titles.

Response 33: We appreciate your valuable comments and we have reformatted the references in the revised manuscript submission. The revision is in References, pages 13-15, lines 559-644.

Reviewer 3 Report

The manuscript deals with an important disease for dairy farming – clinical mastitis caused by Staphylococcus aureus. New lipopeptide antibiotics were synthesized and their antimicrobial activity and toxicity were tested. The most suitable compounds were tested in mice with mastitis induced by Staphylococcus aureus. The English language should be revised because the tense of the verbs was very often wrong. Although the idea is novel and interesting, the manuscript was written in a way which does not support its reading and understanding. First English should be revised to a significant extent and it can be re-submitted and evaluated. Therefore, it cannot be accepted for publication in its current form.

Abstract provides enough information, however, it is not clear how many substances were tested.

Key words: Please, add “mice”.

Introduction

Lines 43-44: “Because they are less likely to develop drug resistance than traditional antibiotics” – Please, provide information why these compounds have lower potential to provoke resistance. Add a maximum of two sentences.

Material and methods

This section should be extended with a better explanation of the methods.

Line 84: “Dissolve the three in sterile PBS to 10 mg/ml concentration and store at -20 °C for backup.” – the style of English in this sentence should be revised, because it is more suitable for a laboratory protocol and for an article.

Line 86 “Determination of MICs”: Please, cite the guideline which was applied for MIC determination.

Line 91: “bacterial solution” should be replaced by “bacterial suspension”. Solution is not a suitable term in the case with bacteria, they are not dissolved.

Lines 95-110, Scanning electron microscope (SEM) assay: The information in this section is very poorly written.

Were the data statistically evaluated? There is no information about that. If statistical analysis was not performed so far, it has to be done for the revised version.

Results

Line 180: “damage is the most obvious” – please, explain exactly how the bacteria were damaged.

The caption of Figures 4 and 5 is not clear to which figures were related to. Please revise, where is figure 5?

Lines 225-226: “treated group were relatively more mentally active than those in the untreated group.” – “more mentally” is not a suitable term. Please, revise!

First, the RT-PCR analysis was not well described in the material and methods. The conditions, the reagents and primers were not mentioned, then the information in the results is actually missing, apart from a single sentence in the last part of this section.

Discussion

Lines 293-298: Please, provide references.

Lines 302-322: Please, compare the antibacterial activity of the new compounds with the existing lipopeptides with antibacterial activity.

The discussion contains some repetitions of the results (see line 332, for example). Please, revise.

The authors also have to explain why an antibiotic from cephalosporins group (cefotaxime) has been chosen.

The effect of the tested compounds on cytokines expression should be better discussed. Please, discuss them in their relationship, especially IL-10 and IL-12.

Conclusion

The conclusion is not precise and overestimates the achievements of the study. It has to be revised in order to properly reflect the results.

The reference list does not follow the style of MDPI journals.

Author Response

Response to Reviewer 1 Comments

Point 1: The manuscript deals with an important disease for dairy farming – clinical mastitis caused by Staphylococcus aureus. New lipopeptide antibiotics were synthesized and their antimicrobial activity and toxicity were tested. The most suitable compounds were tested in mice with mastitis induced by Staphylococcus aureus. The English language should be revised because the tense of the verbs was very often wrong. Although the idea is novel and interesting, the manuscript was written in a way which does not support its reading and understanding. First English should be revised to a significant extent and it can be re-submitted and evaluated. Therefore, it cannot be accepted for publication in its current form.

Response 1: Thank you for your positive comments on the article. We apologize for the problems with manuscript writing. The newly submitted manuscript has been edited in English through the MPDI language retouching system. We hope that you will accept the revised manuscript.

Point 2: Abstract provides enough information, however, it is not clear how many substances were tested.

Response 2: Thank you for your valuable comments. We apologize for the unclear description of the test substance. We have added to this section in the revised manuscript. The revision is in the abstract, page 1, line 16.

Point 3: Key words: Please, add “mice”.

Response 3: We are very grateful for your valuable suggestions. We have added the word "mice" to the keyword section of the revised manuscript. The revision is in the Key words, page 1, line 28.

Point 4: Introduction. Lines 43-44: “Because they are less likely to develop drug resistance than traditional antibiotics” – Please, provide information why these compounds have lower potential to provoke resistance. Add a maximum of two sentences.

Response 4: We appreciate your valuable comments. Lipopeptides are less likely to be resistant because they generally destroy cell membranes, causing damage that is difficult to repair. At the same time, traditional antibiotics act on enzymes or DNA, so they are less likely to be resistant than traditional antibiotics. And we have added this part and related references in the revised manuscript. Revised in Introduction, p.1-2, lines 43-46.

Point 5: Material and methods. This section should be extended with a better explanation of the methods.

Response 5: Thank you for your valuable comments. We have rewritten the Materials and Methods section and hope that the revised manuscript will better explain these methods. Revisions are in Materials and Methods, pages 2-5, lines 68-196.

Point 6: Line 84: “Dissolve the three in sterile PBS to 10 mg/ml concentration and store at -20 °C for backup.” – the style of English in this sentence should be revised, because it is more suitable for a laboratory protocol and for an article.

Response 6: Thank you for your valuable comments. We have revised the "Dissolve the three in sterile PBS to 10 mg/ml concentration and store at -20 °C for backup." to "The lipopeptides were prepared in sterile PBS with a final concentration of 10 mg/ml storage solution and stored at -80°C for backup. "Revisions are in Materials and Methods, pages 2, lines 92-94.

Point 7: Line 86 “Determination of MICs”: Please, cite the guideline which was applied for MIC determination.

Response 7: Thanks to your valuable comments, we have cited references for MIC determination in the revised manuscript. The revision is in Materials and Methods, page 3, lines 97-99.

Point 8: Line 91: “bacterial solution” should be replaced by “bacterial suspension”. Solution is not a suitable term in the case with bacteria, they are not dissolved.

Response 8: We apologize for our carelessness and we have corrected "bacterial solution" to "bacterial suspension". Thank you for your correction! Correction in Materials and Methods, page 3, line 102.

Point 9: Lines 95-110, Scanning electron microscope (SEM) assay: The information in this section is very poorly written.

Response 9: Thank you for your valuable comments. We have rewritten scanning electron microscopy analysis and hope that the revised description will meet your expectations. Revision in Materials and Methods, page 3, lines 106-116.

Point 10: Were the data statistically evaluated? There is no information about that. If statistical analysis was not performed so far, it has to be done for the revised version.

Response 10: All data results in the experiments were statistically analyzed. We apologize for the lack of detail in the description in the previous manuscript. In the revised manuscript, we have described this section in more detail. The revision is in Materials and Methods, p. 5, lines 191-196.

Point 11: Results. Line 180: “damage is the most obvious” – please, explain exactly how the bacteria were damaged.

Response 11: Thank you for your valuable comments. In the revised manuscript we have provided a more detailed description of the damage done to the bacteria, and we hope that the rewriting will better convey the ideas in the article. Revisions are in Results, page 6, lines 215-222.

Point 12: The caption of Figures 4 and 5 is not clear to which figures were related to. Please revise, where is figure 5?

Response 12: We appreciate the reviewer's reminder. We have re-uploaded this image in the revised manuscript, and it is marked as Figure 6 in revised manuscript. Results, page 10, lines 386-402.

Point 13: Lines 225-226: “treated group were relatively more mentally active than those in the untreated group.” – “more mentally” is not a suitable term. Please, revise!

Response 13: Thank you for your valuable comments. We have revised the "the mice in the treated group were relatively more mentally active than those in the untreated group" to "The results showed that mice in the untreated mastitis group were unresponsive, with a disheveled coat and red and swollen mammary gland area. The mice in the lipopeptide and antibiotic-treated groups were responsive, with a neat coat and no apparent redness and swelling in the mammary glands." Revisions are in Result, pages 7, lines 271-274.

Point 14: First, the RT-PCR analysis was not well described in the material and methods. The conditions, the reagents and primers were not mentioned, then the information in the results is actually missing, apart from a single sentence in the last part of this section.

Response 14: Thanks to your valuable comments, we have added and improved the RT-PCR analysis in the revised manuscript. The revision is in Materials and Methods, page 4, lines 168-180.

Point 15: Discussion. Lines 293-298: Please, provide references.

Response 15: Thanks to your comments. We have cited the relevant references in the revised manuscript and they are marked as references 5 and 21 in the revised manuscript. Revision in Discussion, p. 11, line 441.

Point 16: Lines 302-322: Please, compare the antibacterial activity of the new compounds with the existing lipopeptides with antibacterial activity.

Response 16: Thank you for your comments. We have included in the revised manuscript a comparison of the antimicrobial activity of the lipopeptide synthesized in this study with a lipopeptide (C16KGGK) that has been reported to have antimicrobial activity. Revision in Discussion, p. 11, lines 458-469.

Point 17: The discussion contains some repetitions of the results (see line 332, for example). Please, revise.

Response 17: Thank you for your comments. In the revised manuscript, we have removed duplicate content to make the article more concise and clear. Revision in Discussion, p. 11-12, lines 483-487.

Point 18: The authors also have to explain why an antibiotic from cephalosporins group (cefotaxime) has been chosen.

Response 18: Thank you very much for your valuable comments. Due to an oversight on our part, it was not clearly stated in the original manuscript. Cephalosporins are commonly used locally for the treatment of mastitis in dairy cows. Furthermore, in our previous study, we tested the drug sensitivity of local isolates of Staphylococcus aureus (including GS1311 involved in this study), causing mastitis in dairy cows. The results showed that GS1311 was sensitive to cefotaxime (results not published). Therefore, we chose cefotaxime as a control group for antibiotic treatment to compare the effectiveness of conventional antibiotics and synthetic lipopeptides in mastitis treatment. This section and related references have been added to our revised manuscript. Modified in Results, p. 7, lines 266-271.

Point 19: The effect of the tested compounds on cytokines expression should be better discussed. Please, discuss them in their relationship, especially IL-10 and IL-12.

Response 19: Thank you for your valuable comments. In the revised manuscript, we have added a relevant discussion on the relationship between cytokine expression. The revision is in Discussion, p. 12, lines 501-530.

Point 20: Conclusion. The conclusion is not precise and overestimates the achievements of the study. It has to be revised in order to properly reflect the results.

Response 20: Thanks to your valuable comments. We have revised the conclusion section in the revised manuscript to more accurately reflect the study results. Revision in Conclusion, page 12, lines 533-535.

Point 21: The reference list does not follow the style of MDPI journals.

Response 21: We appreciate your valuable comments and we have reformatted the references in the revised manuscript submission. The revision is in References, pages 13-15, lines 559-644.

Round 2

Reviewer 2 Report

Peng et al.:  Synthetic cationic lipopeptide can effectively treat mouse mastitis caused by Staphylococcus aureus

                      3-11-23

Mastitis due to Staphylococcus aureus in dairy cows is common and costly as well as increasingly difficult to treat.  This follows the widespread use of traditional antibiotics and development of drug-resistant strains of bacteria.  Lipopeptide antibiotics have become increasingly important to address this gap in effective antibiotics.  The authors designed and synthesized three cationic lipopeptides with palmitic acid, all of which contained two positive charges and all with dextral amino acids.  Scanning electron microscopy and MIC determinations were used to evaluate activity against S. aureus.  The safe concentration was estimated using mouse erythrocyte susceptibility and CCK8 cytotoxicity tests.  Then, a lipopeptide with bacterial activity and appropriate safety concentrations was tested in a mouse mastitis model.  The authors concluded that one of the three candidates with high activity was found to be able to treat S. aureus mastitis, as tested using the mouse model.  Finally, the authors conclude that their research can serve as a starting point for research into effective antibiotics for mastitis in dairy cows. 

This is a revision of a manuscript on research on novel lipopeptide antibiotics for treatment of bovine mastitis, intended to provide alternatives to traditional antibiotics.  This manuscript addresses and presents new information on this important topic.  The authors responded to many of the issues in the prior review.   Comments to consider at this time include:

Introduction, p. 1, l. 37:  It would be preferable to remove the word “frequently” here.  In reality, S. aureus can cause those effects, but probably not “frequently.”

Introduction, p. 1, check the sentence on ll. 45-46:  Doesn’t it make more sense to state that lipopeptides can be useful additions to the broad class of antibiotics, including traditional ones? 

Introduction, p. 2, ll. 62-67:  The last sentence clearly states the aim of the study.  It needs to be connected more smoothly to the preceding part of the paragraph.

Materials & Methods (M&M), p. 3, l. 124:  The period in this sentence is out-of-place.

M&M, p. 3, ll. 134-135:  The 2 sentences here both contain the phrase “was further measured.”   This could be remedied by combining the 2 sentences something like the example that follows:  “The solution was further measured using a SpectraMax 13X (Molecular Devices) at an optical density (OD) of 450 nm.” 

M&M, p. 4, l. 161:  Section 2.9 has a heading that refers to “breast tissue.”  This should appropriately be referred to as “mammary gland tissue” or something similar.

M&M, p. 5, l. 188:  Should the asterisk (*) in Table also be placed before the footnote “m” means “mouse”?

M&M, p. 5, l. 196:  Why was significance considered at the P < 0.01 level rather than the traditional P < 0.05?

Results, p. 6, ll. 207-209:  Where is Figure 1?

Results, p. 6, l. 217:  Could you explain “cell crumpling” or use another word to describe this?

Results, p. 6, ll. 220 and 222:  You reference “bacteriophages” here.  Maybe I am wrong or missing something, but I don’t think this is correct.

Results, p. 6, ll. 228-230, Figure 2:  The Figure should be able to stand alone.  Please note or explain the arrows in the figure.

Results, p. 7, ll. 233-234:  Here you state that:  “The detection of hemolytic activity can well verify drug safety.”  Reading the reference provided, it would appear that hemolytic activity alone cannot fully verify drug safety.  It would be more appropriate to state that hemolytic activity can be one indication of drug safety.

Results, p.  8, ll. 277-278:  You write that “Notably, in the two different infection groups, we found the mammary tissue recovery was better in the lipopeptide-treated group than in the antibiotic-treated group.”   Please tell the reader how you determined this, like with an objective scale—or state more conservatively, such as “it was our visual assessment.” or “Our assessment was that the …”

Results, p.  9, ll. 359-371:  You make statements here about many “significant” findings.  Please clarify which were statistically assessed and, when so, give p values.

Results, Figure 1, p. 10, l. 401:  For A and C:   You don’t indicate on the graphs to the far left of the first columns were bacterial levels.

Discussion, p. 10, l. 410:  There is a repeat of the word “the” here.

Discussion, p.  11, l. 461:  What is “gold glucose”?

Discussion, p. 11 , ll. 463-464:  What do you mean by “Extensive range of safe action concentrations.”

Discussion, p. 11, l. 467:   When you refer to the “glands of these 2 species” do you mean cattle and mice?   If so, you might make this clear.

Discussion, p. 11, ll. 478-484:  Here you make 2 statements that are not clear:  1.  “At present, there are two diagnostic methods for mastitis, qualitative and quantitative.”  It is not clear how this fits into the paragraph.  2.  “Firstly, we observed the ocular pathological changes in the mammary glands of mice in different treatment groups after the treatment experiment.”  It is not clear what you mean when you refer to “ocular pathological changes.”

Discussion, p. 12, ll. 485-486:  “Even in terms of phenotypic changes, the therapeutic effect of C16dKdK was superior to that of cefoxatime.”  Please see comments for Results, p. 8, ll. 277-278.

References, p. 13, ll. 577-578:  Citation in text does not agree with reference.   Please check.

                      p. 13, l. 14:  Reference #14 just has author initials, not complete name.

Author Response

Response to Reviewer 2 Comments

Point 1: Introduction, p. 1, l. 37: It would be preferable to remove the word “frequently” here. In reality, S. aureus can cause those effects, but probably not “frequently.”

Response 1: Thank you for your correction. In the revised manuscript, we have removed "frequently". The specific change is in the Introduction page 1, line 37.

Point 2: Introduction, p. 1, check the sentence on ll. 45-46: Doesn’t it make more sense to state that lipopeptides can be useful additions to the broad class of antibiotics, including traditional ones? 

Response 2: Thank you for your valuable questions. Due to our oversight this section was described incorrectly, and we have rewritten this section in the revised manuscript, with the changes located at the Introduction pages 1-2, lines 43-46.

Point 3: Introduction, p. 2, ll. 62-67: The last sentence clearly states the aim of the study. It needs to be connected more smoothly to the preceding part of the paragraph.

Response3: Thank you for your valuable comments. We have rewritten this section in the revised manuscript to make the natural paragraph flow smoothly and naturally. The specific change is in the Introduction page 2, lines 66-71.

Point 4: Materials & Methods (M&M), p. 3, l. 124: The period in this sentence is out-of-place.

Response 4: Thank you for your correction. In the revised manuscript, we have revised the formatting of the punctuation. Revisions are in Materials and Methods, page 3, line 130.

Point 5: M&M, p. 3, ll. 134-135: The 2 sentences here both contain the phrase “was further measured.” Th is could be remedied by combining the 2 sentences something like the example that follows: “The solution was further measured using a SpectraMax 13X (Molecular Devices) at an optical density (OD) of 450 nm.”

Response 5: Thank you for your correction. In the revised manuscript, we have combined the two sentences as you suggested and revised them to "The solution was further measured using a SpectraMax 13X (Molecular Devices) at an optical density (OD) of 450 nm." Revisions are in Materials and Methods, pages 3-4, lines 140-141.

Point 6: M&M, p. 4, l. 161: Section 2.9 has a heading that refers to “breast tissue.” This should appropriately be referred to as “mammary gland tissue” or something similar.

Response 6: Thank you for your correction. In the revised manuscript, we have changed "breast tissue" to "mammary gland tissue". Revisions are in Materials and Methods, page 4, line 166.

Point 7: M&M, p. 5, l. 188: Should the asterisk (*) in Table also be placed before the footnote “m” means “mouse”?

Response 7: Thank you for your correction. In the revised manuscript, we have added asterisk (*) to the top of the table notes. Revisions are in Materials and Methods, page 5, lines 193.

Point 8: M&M, p. 5, l. 196: Why was significance considered at the P < 0.01 level rather than the traditional P < 0.05?

Response 8: Thank you for your valuable questions. Generally, P < 0.05 is considered statistically different, and P < 0.01 is considered statistically significantly different, so we used P<0.01.

Point 9: Results, p. 6, ll. 207-209: Where is Figure 1?

Response 9: We apologize for the loss of Figure 1 in the manuscript, but this may be due to the submission system, as there was a problem with some of the images being lost in the first submission. To avoid this problem, we uploaded a Word version of the manuscript along with a PDF version of the manuscript to ensure the integrity of the manuscript. Revisions are in Results, page 5, line 210.

Point 10: Results, p. 6, l. 217: Could you explain “cell crumpling” or use another word to describe this?

Response 10: Thank you for your question. The term "cell crumpling" refers to the state in which the cell shape becomes less full compared to normal cells after lipopeptide treatment.

Point 11: Results, p. 6, ll. 220 and 222: You reference “bacteriophages” here. Maybe I am wrong or missing something, but I don’t think this is correct.

Response 11: Thank you very much for your question. We were trying to convey the word bacterial cell, but we used the wrong word. In the revised manuscript we have corrected "bacteriophages" to "bacterial cell". Revisions are in Results, page 6, lines 224, 226.

Point 12: Results, p. 6, ll. 228-230, Figure 2: The Figure should be able to stand alone. Please note or explain the arrows in the figure.

Response 12: Thank you very much for your valuable comments. I am sorry I did not understand what "The Figure should be able to stand alone" means. Figure 2 shows the electron microscopic observation of the structure of Staphylococcus aureus after treatment with lipopeptides. We have modified the meaning of the arrows in Figure 2. The arrow in the C16dKdK treated ATCC25923 infection group indicates an apparent crack on the cell surface; the arrow in the C16dKdK treated GS1311 infection group indicates severe crumpling;  the arrows in the C16dKGdK and C16dKGGdK treated ATCC25923 infected groups indicate roughness and small cracks on the surface of the cell, which are less severe than those in the C16dKdK treated ATCC25923 infected group; the arrows in the C16dKGdK and C16dKGGdK treated GS1311 infected groups indicate different degrees of crumpling of the bacterium, which are less severe than those in the C16dKdK treated GS1311 infected group. Revisions are in Results, page 6, lines 231-236.

Point 13: Results, p. 7, ll. 233-234: Here you state that: “The detection of hemolytic activity can well verify drug safety.” Reading the reference provided, it would appear that hemolytic activity alone cannot fully verify drug safety. It would be more appropriate to state that hemolytic activity can be one indication of drug safety.

Response 13: We appreciate your valuable comments. In the revised manuscript, we have changed "The detection of hemolytic activity can well verify drug safety." to "Hemolytic activity can be one indication of drug safety. Revisions are in Results, page 7, lines 238-239.

Point 14: Results, p.  8, ll. 277-278: You write that “Notably, in the two different infection groups, we found the mammary tissue recovery was better in the lipopeptide-treated group than in the antibiotic-treated group.”  Please tell the reader how you determined this, like with an objective scale—or state more conservatively, such as “it was our visual assessment.” or “Our assessment was that the …”

Response 14: We appreciate your valuable comments. In the revised manuscript, we have rewritten this part, and we hope that the revision will make the article more rigorous. Revisions are in Results, page8, lines 290-296.

Point 15: Results, p.  9, ll. 359-371: You make statements here about many “significant” findings. Please clarify which were statistically assessed and, when so, give p values.

Response 15: Thank you for your valuable comments. We have revised the content of this section. Regarding the calculation method and determination criteria of p-value, we wrote in the material methods section of the article manuscript. Revisions are in Materials and Methods, page 5, lines 194-200.

Point 16: Results, Figure 1, p. 10, l. 401: For A and C: You don’t indicate on the graphs to the far left of the first columns were bacterial levels.

Response 16: Thank you for your corrections. We apologize for our carelessness. In the revised manuscript we indicate on the graphs to the far left of the first columns were bacterial levels. Revisions are in Results, page 10, line 352.

Point 17: Discussion, p. 10, l. 410: There is a repeat of the word “the” here.

Response 17: Thank you for your corrections. We apologize for our carelessness, and we have removed the word "the" in the revised manuscript. Revision in Discussion, page 10, line 363.

Point 18: Discussion, p.  11, l. 461: What is “gold glucose”?

Response 18: Thank you for your correction. We apologize for the incorrect word. In the revised manuscript "gold glucose" has been corrected to "S. aureus". Revision in Discussion, page 11, lines 411,416.

Point 19: Discussion, p. 11 , ll. 463-464: What do you mean by “Extensive range of safe action concentrations.”

Response 19: We appreciate your valuable question. We apologize for the inaccurate description in the manuscript, and we have rewritten this section in the revised manuscript. Revision in Discussion, page 11, lines 419-420.

Point 20: Discussion, p. 11, l. 467: When you refer to the “glands of these 2 species” do you mean cattle and mice?   If so, you might make this clear.

Response 20: We appreciate your valuable question. We apologize for the lack of clarity in our description of this issue. We have rewritten this content in the revised manuscript and hope that the revised content will clearly express the content of the article. Revision in Discussion, page 11, line 423.

Point 21: Discussion, p. 11, ll. 478-484: Here you make 2 statements that are not clear: 1. “At present, there are two diagnostic methods for mastitis, qualitative and quantitative.” It is not clear how this fits into the paragraph. 

Response 21: Thank you very much for your valuable comments. In the revised manuscript, we have added a more detailed description of the section "Relevant tests were classified qualitatively and quantitatively". Revision in Discussion, page 12, lines 437-439.

Point 22: 2. “Firstly, we observed the ocular pathological changes in the mammary glands of mice in different treatment groups after the treatment experiment.” It is not clear what you mean when you refer to “ocular pathological changes.”

Response 22: Thank you very much for your valuable comments. “ocular pathological changes.” means “visual assessment about pathological changes”. And we have corrected it in the revised manuscript. Revision in Discussion, page 12, line 440.

Point 23: Discussion, p. 12, ll. 485-486: “Even in terms of phenotypic changes, the therapeutic effect of C16dKdK was superior to that of cefoxatime.” Please see comments for Results, p. 8, ll. 277-278.

Response 23: Thank you for your valuable comments. A detailed description of the determination of this result has been added to the revised manuscript. Revision in Discussion, page 12, line 444.

Point 25: p. 13, l. 14: Reference #14 just has author initials, not complete name.

Response 25: We thank you for your reference. In the revised manuscript, we have refined the names of the authors of the references. The revision is in References, page 14, lines 539.

Reviewer 3 Report

The authors considered all the remarks and improved significantly the presentation of the conducted experiments. The manuscript can be accepted for publication.

Two editorial changes are necessary:

Line 64: “The length of the D-lysine peptide chain influences the antibacterial activity of ATCC25923 and GS1311.” Should be “The length of the D-lysine peptide chain influences the antibacterial activity against S. aureus ATCC25923 and S. aureus GS1311.”

Line 140: “strains of ATCC25923 and GS1311” – please, use S. aureus before the numbers of strains. Follow this style through the whole manuscript.

Author Response

Response to Reviewer 3 Comments

Point 1: Line 64: “The length of the D-lysine peptide chain influences the antibacterial activity of ATCC25923 and GS1311.” Should be “The length of the D-lysine peptide chain influences the antibacterial activity against S. aureus ATCC25923 and S. aureus GS1311.”

Response 1: Thank you for your valuable comments. We have rewritten this section in the revised manuscript to make the natural paragraph flow smoothly and naturally. The specific change is in the Introduction page 2, lines 66-71.

Point 2: Line 140: “strains of ATCC25923 and GS1311” – please, use S. aureus before the numbers of strains. Follow this style through the whole manuscript.

Response 2: Thank you for your corrections. We have added " S. aureus" before the representative numbers indicating the corresponding strains throughout the revised manuscript. Revisions are in the full text, lines76, 114, 146, 154, 155 220, 221, 223, 225, 228, 280, 288, 289, 328, 411, 416, 433, 467 and 468.